# Detection of hidden antibiotic resistance through real-time genomics

Ela Sauerborn ●[1,2,3,4], Nancy Carolina Corredor ●[4], Tim Reska ●[1,2,3], Albert Perlas[1,2], Samir Vargas da Fonseca Atum ●[1,2,5,6], Nick Goldman ●[7], Nina Wantia[4], Clarissa Prazeres da Costa ●[4,8,9], Ebenezer Foster-Nyarko[10] & Lara Urban ●[1,2,3] ✉

Real-time genomics through nanopore sequencing holds the promise of fast antibiotic resistance prediction directly in the clinical setting. However, concerns about the accuracy of genomics-based resistance predictions persist, particularly when compared to traditional, clinically established diagnostic methods. Here, we leverage the case of a multi-drug resistant *Klebsiella pneumoniae* infection to demonstrate how real-time genomics can enhance the accuracy of antibiotic resistance profiling in complex infection scenarios. Our results show that unlike established diagnostics, nanopore sequencing data analysis can accurately detect low-abundance plasmid-mediated resistance, which often remains undetected by conventional methods. This capability has direct implications for clinical practice, where such "hidden" resistance profiles can critically influence treatment decisions. Consequently, the rapid, in situ application of real-time genomics holds significant promise for improving clinical decision-making and patient outcomes.

The World Health Organization has declared antibiotic resistance one of the ten most severe global health threats[1], with resistant infections leading to higher mortality and morbidity due to delayed or inappropriate therapy[2]. The rapid and accurate identification of resistant bacterial pathogens could facilitate the earlier administration of appropriate therapy, decreasing the mortality rate and infection- and treatment-related morbidity[3].

Real-time genomics, powered by nanopore sequencing technology, offers the potential to expedite pathogen identification and antibiotic resistance profiling directly within clinical settings[4,5]. The portability of this technology, coupled with its capability for real-time analysis, enables cost-efficient adaptive applications, where as much

genomic data as needed can be directly obtained on-site to reach minimum certainty thresholds for making timely and clinically relevant predictions[6]. However, for real-time genomics to be integrated into routine clinical practice, its accuracy in predicting antibiotic resistance must be directly compared with that of established diagnostic approaches[7]. While several proof-of-concept studies have showcased the feasibility of using nanopore sequencing for rapid infectious disease diagnostics in clinical settings[5–9], it remains to be proven that real-time genomics can outperform established diagnostics in detecting clinically relevant resistance.

Nanopore sequencing's capability to produce long reads can be leveraged to create highly accurate, near-complete genome

[1]Helmholtz AI, Helmholtz Zentrum Muenchen, Neuherberg, Germany. [2]Helmholtz Pioneer Campus, Helmholtz Zentrum Muenchen, Neuherberg, Germany. [3]Technical University of Munich (TUM), School of Life Sciences, Freising, Germany. [4]Institute of Medical Microbiology, Immunology and Hygiene, TUM School of Medicine and Health, TUM School of Medicine and Health, Technical University of Munich, Munich, Germany. [5]Departamento de Química Fundamental, Instituto de Química, Universidade de São Paulo, São Paulo, Brazil. [6]Departamento de Bioquímica, Instituto de Química, Universidade de São Paulo, São Paulo, Brazil. [7]European Molecular Biology Laboratory, European Bioinformatics Institute (EMBL-EBI), Wellcome Genome Campus, Cambridge, UK. [8]Center for Global Health, TUM School of Medicine and Health, Technical University of Munich, Munich, Germany. [9]German Center for Infection Research (DZIF), partner site Munich, Munich, Germany. [10]Department of Infection Biology, London School of Hygiene & Tropical Medicine, Keppel Street, London, UK. ✉e-mail: lara.h.urban@gmail.com

assemblies for strain-level identification and de novo detection of bacterial pathogens and their antibiotic resistance profiles[10]. This is particularly pertinent for complex infections, where clinically established methods for taxonomic bacterial identification (e.g., MALDI-TOF mass spectrometry) and resistance profiling (e.g., VITEK2) might lack resolution, but where rapid, targeted therapy can be particularly beneficial for patient outcome.

Here, we show the power of in situ real-time genomics in a clinical setting through the example of a *Klebsiella pneumoniae* infection for which real-time genomics-based resistance predictions—in contrast to clinically established diagnostics—could identify a novel antibiotic resistance gene variant located on low-abundance plasmids. This finding has significant implications for clinical decision-making and potentially for patient outcomes, illustrating the transformative potential of integrating real-time genomic analysis into clinical practice.

## Results

We conducted a comparative analysis of the performance between clinically established diagnostics and real-time genomics-based predictions using bacterial isolates from the same infection case. Our established diagnostic methods included MALDI-TOF MS for taxonomic bacterial identification and VITEK2 for antibiotic resistance profiling (Methods; Fig. 1). The case study involved an immunocompromised patient at the University Hospital rechts der Isar in Germany, who presented with a fever and was initially treated with the carbapenem antibiotic Meropenem.

This comparative approach allowed us to directly assess the accuracy and speed of real-time genomic predictions against traditional methods in a real-world clinical scenario, highlighting the potential advantages of genomic technologies in rapid and accurate pathogen identification and resistance prediction.

### Clinically established diagnostics

Initially, an endotracheal aspirate sample was collected ("pre-treatment" sample), from which clinically established diagnostics identified a carbapenem-resistant *K. pneumoniae* isolate, with *K. pneumoniae* carbapenemase (KPC) as the putative resistance-conferring mechanism (Methods; Fig. 2). KPCs can hydrolyze a variety of beta-lactam antibiotics, including carbapenems[11]. Consequently, Ceftazidime-Avibactam (CAZ-AVI) is recommended as a treatment option[12] due to its efficacy against such resistant strains; in this specific case, following the diagnostics that also predicted CAZ-AVI susceptibility, the treatment was promptly switched from Meropenem to CAZ-AVI (Methods; Fig. 2). After initial clinical improvement, the patient's condition deteriorated under CAZ-AVI therapy, and a subsequent blood culture taken from the patient ("post-treatment" sample) grew a *K. pneumoniae* isolate which now showed restored in-vitro carbapenem susceptibility but CAZ-AVI resistance (Fig. 2). Notably, while new KPCs variants that confer resistance to CAZ-AVI have been documented[13–15], the clinically established diagnostics failed to detect any carbapenemase in the post-treatment isolate. Consequently, Meropenem was reintroduced as part of the treatment regimen. Despite these measures, the patient passed away shortly afterwards (Fig. 2).

### Real-time genomics diagnostics

To explore the potential of real-time genomics in antibiotic resistance prediction, we simulated its application for this clinical case as follows. We applied rapid nanopore shotgun sequencing (Oxford Nanopore Technologies) on both the pre-and post-treatment *K. pneumoniae* bacterial isolates (Methods) using the portable Mk1b sequencing device and rapid barcoding library preparation (Fig. 2; Supplementary Table 1; Methods). We processed the raw nanopore data through high-accuracy basecalling, de novo genome assembly, species identification, and antibiotic resistance prediction using EPI2ME's Antimicrobial

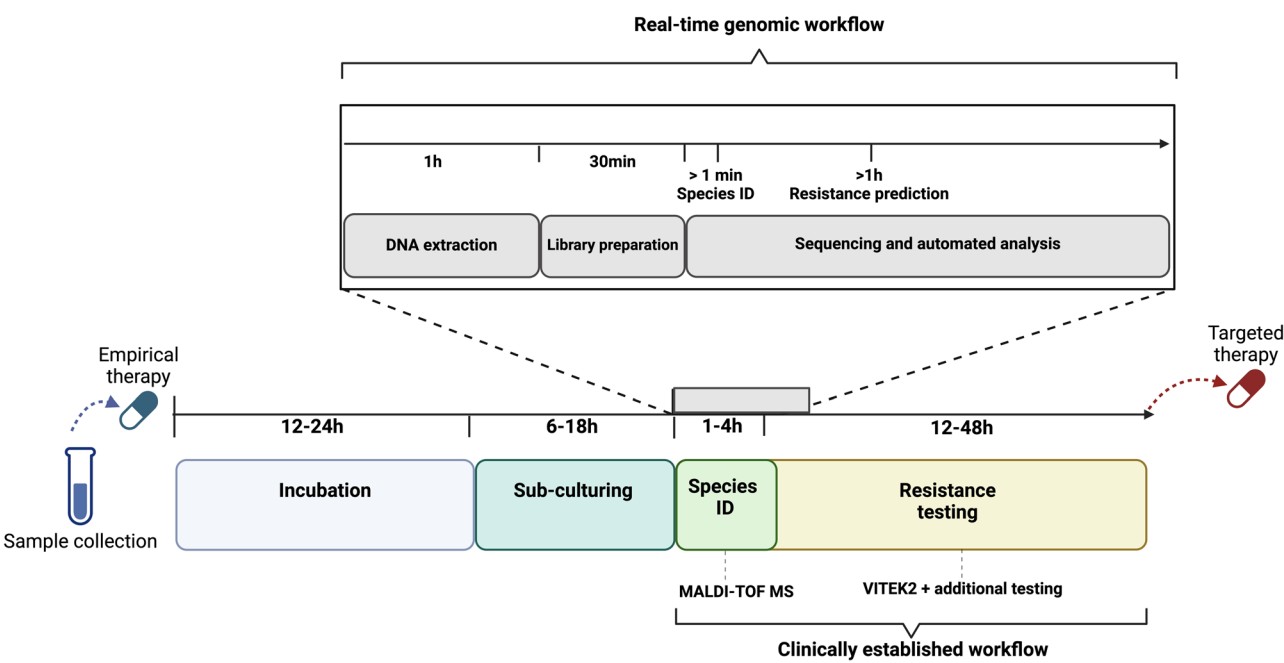

Workflow comparing a culture-based and a nanopore-based approach.
ID: Identification, AST: antibiotic susceptibility testing, MALDI-TOF MS: Matrix-assisted laser desorption ionization time-of-flight mass spectrometry.

**Fig. 1 | Workflow overview of real-time genomic (*top*) and clinically established (*bottom*) diagnostic approaches for pathogen species identification and antibiotic resistance profiling.** After incubation and primary pathogen identification, pure bacterial isolates are recovered through sub-culturing, followed by pathogen and resistance profiling of the isolates. While the clinically established workflow can take up to 52 h after subculturing, the real-time genomic workflow delivers the first data after ~1.5 h and can be employed in an adaptive manner to create the necessary amount of data. Created with Biorender.com.

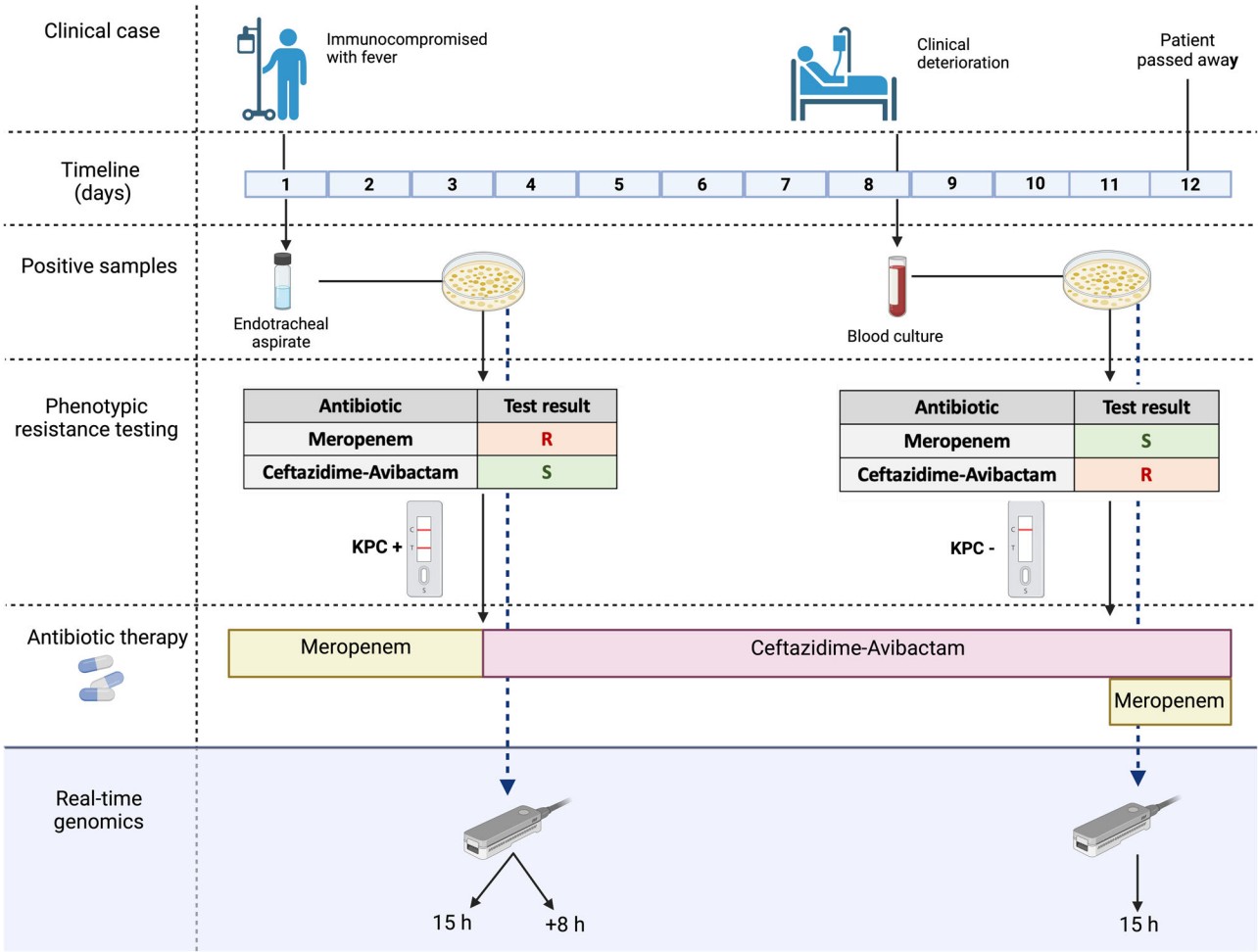

**Fig. 2 | Overview of the course and treatment of the infection case.** The patient was firstly treated with Meropenem. *K. pneumoniae* bacterial isolates of the first positive patient sample (endotracheal aspirate) were subjected to VITEK2 for general resistance testing and additional tests for CAZ-AVI resistance and KPC detection (R resistant, S susceptible; KPC +/-: absence or presence of KPC; Methods); the diagnostics led to a change in the antibiotic treatment to CAZ-AVI after three days. After clinical deterioration, the second isolate (from blood culture) showed reversed antibiotic resistance test results. While Meropenem was subsequently administered, the patient passed away shortly after. After completion of the routine diagnostics, we used real-time genomics to sequence DNA from the pre- and post-CAZ-AVI treatment bacterial isolates using the portable nanopore sequencing device Mk1b (Methods). Both isolates were sequenced for 15 h, and the first isolate was sequenced for another 8 h to simulate the potential of adaptive sequencing in the clinical setting (Methods). Created with Biorender.com.

**Table 1 | Real-time genomic antibiotic resistance predictions from pre- and post-treatment bacterial isolates using EPI2ME's Antimicrobial Resistance protein homolog model**

| Sequencing run | Meropenem | | | | Ceftazidime-Avibactam | | | |
|---|---|---|---|---|---|---|---|---|
| | Prediction | Evidence | Accuracy | CN | Prediction | Evidence | Accuracy | CN |
| Pre-treatment 15 h | R | $bla_{KPC-2}$ | 93.7% | 40 | R? | $bla_{KPC-14}$ | 92.6% | 1 |
| +8 h | R | | 94.4% | 147 | R | | 96.3% | 4 |
| Post-treatment 15 h | S | / | / | / | R | $bla_{KPC-14}$ | 93.4% | 44 |

*R resistant, S susceptible, CN copy-number of respective resistance gene (Methods). Accuracy refers to gene detection accuracy according to EPI2ME's Antimicrobial Resistance protein homolog model (Methods).*

Resistance protein homolog model[16] (Methods; Supplementary Data 1). Our analysis correctly identified *K. pneumoniae* in the pre-and post-treatment isolates as the causative pathogen. In the pre-treatment isolate, we detected accurate (>90%) $bla_{KPC-2}$ gene copies ($n = 40$; Table 1), confirming the Meropenem resistance observed by clinically established diagnostics (Fig. 2)[17]. For the post-treatment isolate, we identified numerous copies of the $bla_{KPC-14}$ ($n = 44$; Table 1), which had previously been recognized as one of the few KPC subtypes that confer CAZ-AVI resistance while potentially restoring in-vitro

carbapenem susceptibility[14]. Hence, the genomics-based resistance prediction not only aligned with the resistance patterns identified by traditional diagnostics but also pinpointed $bla_{KPC-14}$ as the putative resistance-conferring mechanism (Table 1).

Crucially, our real-time genomics approach also detected a single copy of the $bla_{KPC-14}$ resistance gene in the pre-treatment isolate. Although this one copy would not have been sufficient to predict CAZ-AVI resistance initially, this case served as a valuable example to simulate the adaptive nature of real-time genomics applications in the

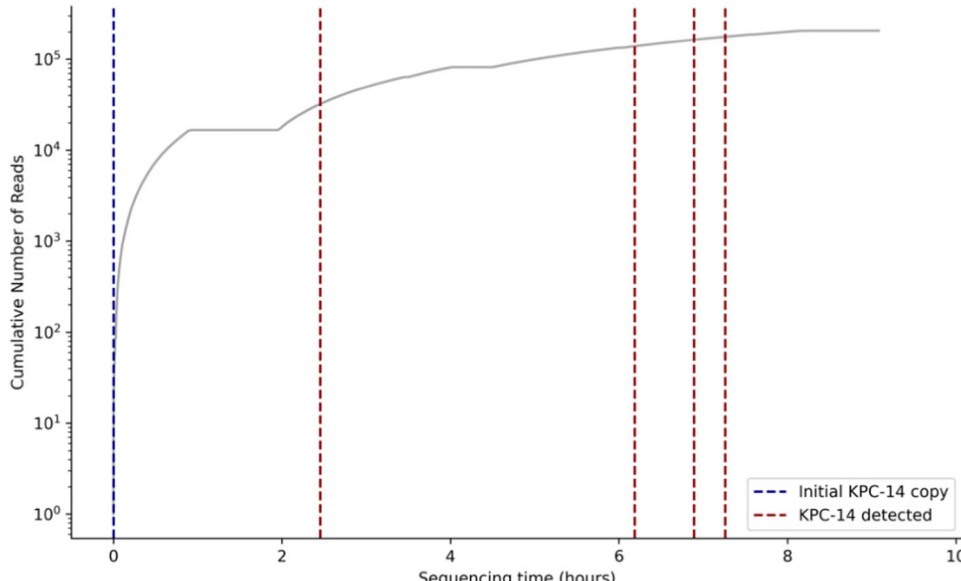

**Fig. 3 | Timeline of $bla_{KPC-14}$ copy number detection during simulation of an adaptive sequencing run in the clinical setting applied to a technical replicate of the pre-treatment bacterial isolate.** The run started with one $bla_{KPC-14}$ copy number which was detected in the first sequencing run, and all additional $bla_{KPC-14}$ copies were detected within the first seven hours of this second sequencing run.

clinic, where a sample can be sequenced as long as necessary to obtain the necessary minimum data for reliable genomics-based predictions. For this purpose, we conducted additional sequencing of a technical replicate of the pre-treatment isolate for another eight hours (Methods; Fig. 2; Supplementary Table 1), which resulted in the identification of four more highly accurate copies of $bla_{KPC-14}$ (Table 1). Remarkably, a second $bla_{KPC-14}$ gene copy would have already been detected after two hours of additional sequencing and would have rapidly indicated the potential of CAZ-AVI resistance (Fig. 3; Methods).

**Epidemiological and functional analyses.** We successfully created de novo assemblies of one complete chromosome and three complete circular plasmids from both the pre- and post-treatment isolate (Fig. 4; Methods). Core-genome multilocus sequence typing (cgMLST; Methods)[18,19] revealed that both isolates were of the emerging high-risk sequence type ST147[20]. Single-linkage clustering analysis identified no close relatives within the 50-allele threshold typically used for cgMLST clustering (Methods), indicating that our bacterial isolates are genetically distinct from globally known ST147 genomes.

Functional annotation of the assembled plasmid genomes revealed that the $bla_{KPC-2}$ (pre-treatment isolate) and $bla_{KPC-14}$ (post-treatment isolate) gene were located on IncN plasmids, which were 99.7%-identical according to sequence alignments (E-score 0, Bit-Score >1.461e + 5; Methods; Fig. 4). Additionally, both IncN plasmids shared key plasmid features (relaxase type: MOBF, mpf type: MPF_T, orit type: MOBF), and were predicted to be conjugative[21,22].

We further inferred a copy-number of three and four for the IncN plasmids relative to the bacterial chromosome in the pre-and post-treatment isolates, respectively (Methods). To assess the changes in the abundance of $bla_{KPC-14}$ gene between pre- and post-treatment isolates, we further normalized the $bla_{KPC-14}$ copy-numbers against the most abundant resistance gene ($bla_{TEM-4}$) detected on the IncN plasmid (Methods). We observed that the normalized abundance of $bla_{KPC-14}$ increased from 0.56% to 26.6% following CAZ-AVI exposure.

Upon submission of the post-treatment bacterial isolate to the German National Reference Center for Gram-negative bacteria, the KPC resistance gene that we initially defined as $bla_{KPC-14}$ was identified as a previously undocumented KPC subtype and subsequently named $bla_{KPC-159}$ (NCBI sequence ID: OQ450354.1). To confirm this, we utilised the BLASTn tool and established that $bla_{KPC-159}$ shows 99.9% similarity in nucleotide sequence (875/876 bases) with the query sequence of $bla_{KPC-14}$ (Methods), thus, leading to the classification of $bla_{KPC-159}$ as $bla_{KPC-14}$.

## Discussion

The application of real-time genomics to this patient's case underscores the considerable potential for using this technology to rapidly and accurately profile complex bacterial infections in the clinical setting. Our findings suggest that the shift in in-vitro antibiotic resistance was likely due to a complex infection involving the same *K. pneumoniae* lineage with a low-abundance $bla_{KPC-14}$-carrying IncN plasmid which became dominant due to its evolutionary selective advantage under CAZ-AVI exposure. We have shown that nanopore sequencing could have unveiled the CAZ-AVI resistance that phenotypic methods failed to detect, thereby influencing the therapeutic approach, such as the early administration of alternative antibiotics or combination therapy[23]. Further, identifying the $bla_{KPC}$ gene as the underlying CAZ-AVI resistance mechanism would have directly informed clinical infection prevention protocols, reducing the risk of between- and within-patient KPC transmission. The fast, adaptive, and in situ nature of antibiotic resistance profiling by nanopore sequencing would have surpassed current clinical practice in accurately informing clinical management.

Importantly, our genomic data also enabled us to trace changes in antibiotic resistance to variations in the copy-number of plasmids within the patient's initial infection. The presence of multiple plasmids in an infection raises public health concerns as it can accelerate the emergence of resistance under selection pressures[24]. This has previously been described as a key factor in the rapid emergence of resistance to the last-resort antibiotic CAZ-AVI under drug pressure[24]. The anticipated increase in antibiotic resistance and the limitations of current diagnostic methods to fully assess complex infections pose significant challenges to effective antibiotic treatment strategies.

While our analysis provides evidence for the added value of real-time genomics for complex bacterial infections, two significant

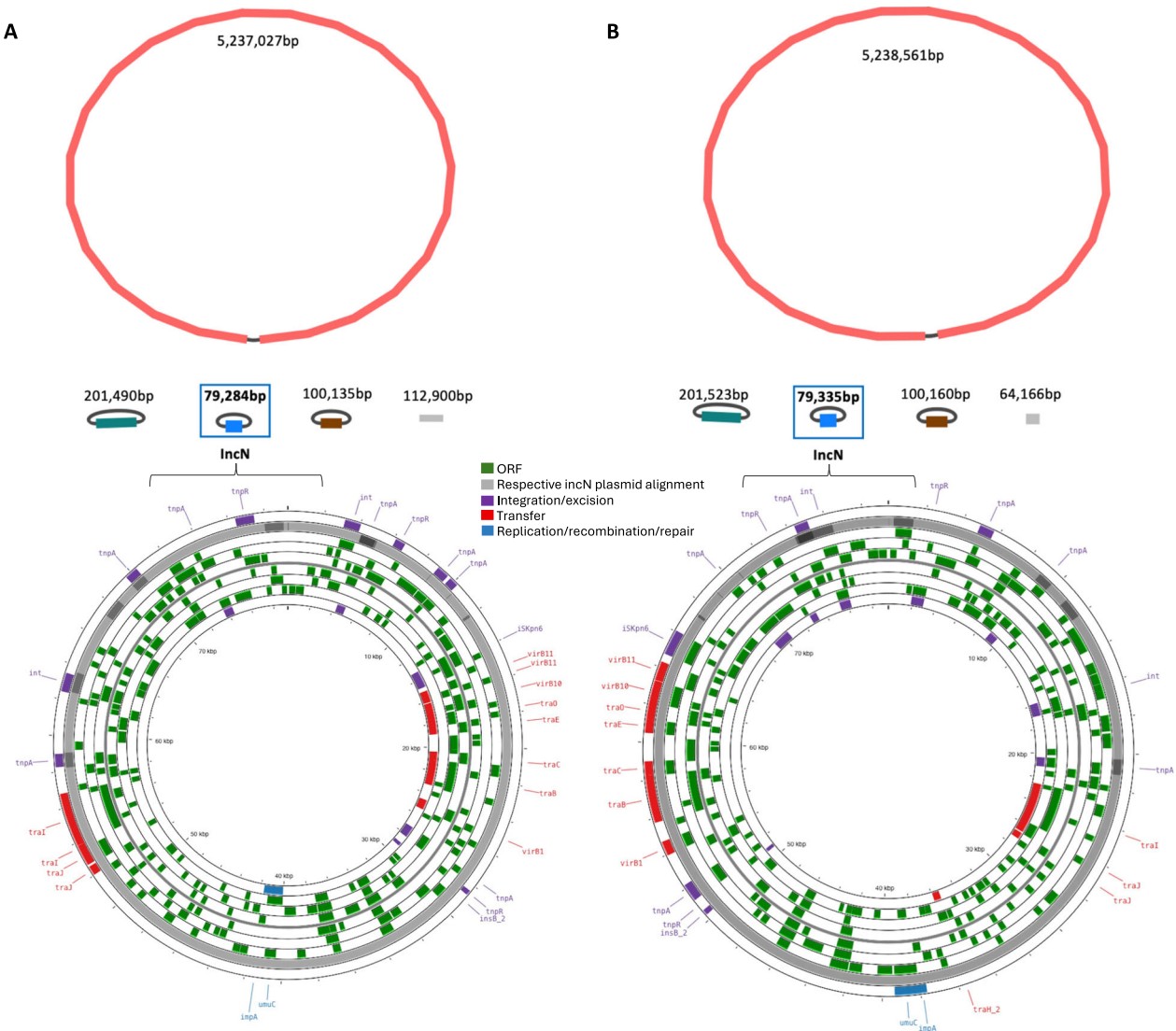

**Fig. 4 | Genome and plasmid de novo assemblies of** A **pre-treatment and** B **post-treatment** *K. pneumoniae* **isolates.** *Top:* The assemblies are annotated by respective contig length; the IncN plasmid is highlighted by the blue square. Additional non-circular contigs are visualized in grey color. Created with Bandage v0.90[34]. *Bottom:* Functional annotation and visualisation of the IncN plasmids highlighting open reading frames (ORF) and plasmid functionality genes (integration/excision, transfer, replication/recombination/transfer) from the mobileOG-db (Methods). The alignment to the respective other IncN plasmid (pre- *vs.* post-treatment sample) is shown in grey color. Created with ProkSee.com[35].

limitations to our work remain. Firstly, focusing on bacterial isolates may have limited our ability to fully understand the microbial diversity of the patient's infection. A metagenomic sequencing approach using direct patient samples, rather than cultured bacteria, could have identified non-culturable organisms that might have influenced the antibiotic resistance profile[7]. Secondly, our analysis was confined to just two bacterial isolates from the same patient over an extended period. A larger sample size and more frequent sampling might have provided more detailed insights into how plasmid selection evolved in response to the antibiotic selection pressure. Unfortunately, due to the retrospective nature of this study, we were unable to adjust the sample size or obtain direct patient samples.

Current clinically established diagnostics are often too slow to promptly inform clinical management and require substantial initial investments in technology[5–7]. Our study showcases how cost-efficient, rapid real-time genomics can outperform established diagnostics in accuracy for predicting antibiotic resistance. Further research is needed to transition from phenotypic resistance testing to genomics-

based predictions fully. However, we already now anticipate great potential for combining the advantages of real-time genomic technology with clinically established approaches for antibiotic resistance profiling in the hospital setting. We further envision that the ongoing improvements in sequencing accuracy and the relatively low investment required for nanopore sequencing technology[25] offer promising prospects for managing complex infection cases worldwide, particularly in low- and middle-income settings where advanced diagnostic equipment may not be readily available.

## Methods

### Clinically established workflow for species identification and antibiotic susceptibility testing

All ethical approval for the following clinical research was given by the ethics committee of the Technical University of Munich, Germany (2023-575-W-NP, 2022-611-S-KH). Informed patient consent was waived as samples were taken under routine diagnostics. This research conforms to the principles of the Helsinki Declaration.

Initially, the clinical samples were plated out on BD® Columbia Blood and MacConkey agar plates (Beck Dickinson GmbH, Heidelberg, Germany) and incubated for approximately 16–24 h. Following subculture, species identification was done from a single colony forming unit (CFU) of pure bacterial isolates using Matrix-assisted laser desorption ionization time-of-flight mass spectrometry instructions (MALDI-TOF MS, Bruker Daltronics GmbH, Leipzig Germany), as per the manufacturer's instructions.

Antibiotic susceptibility testing was performed using VITEK2 (BioMérieux, Marcy l'Etoile, France). For this, up to three bacterial CFUs were transferred to a saline tube to generate a homogenous suspension with a density equivalent to 0.5 McFarland. Subsequently, Minimum Inhibitory Concentrations (MICs in mg/L) of the two isolates were determined using the VITEK2 gram-negative (AST-GN69) card, and the results interpreted according to the European Committee on Antimicrobial Susceptibility Testing (EUCAST) guidelines[26] (Supplementary Table 2).

Given that the AST-GN69 card does not detect the presence of carbapenemases or measure MICs for CAZ-AVI, we additionally performed the following tests. For isolates with antibiotic susceptibility profiles indicative of carbapenem resistance, we identified carbapenemases with a multiplex immunochromatography assay consisting of lateral flow assays (O.K.N.V.I Resist-5[27]). This test detects the presence of Oxa-48, NDM, VIM, IMP-carbapenemases and the most prevalent KPC subtypes (e.g., KPC-2, KPC-3)[28,29]. CAZ-AVI MIC was measured with the Liofilchem® MIC test strip (MTS™) (LIOFILCHEM s.r.l., Roseto degli Abruzzi, Italy), which contains Ceftazidime concentrations ranging from 0,016–256 μg/ml with a fixed Avibactam concentration of 4 μg/ml. Similar to the VITEK2 MIC measurements, up to three bacterial CFUs were transferred to a sterile saline tube to form a homogenised suspension with a density equivalent to 0.5 McFarland. This suspension was then plated out on a BD® Muller Hinton Agar using sterile cotton-tipped swabs and incubated with a CAZ-AVI Liofilchem® MTS™ for 16 h. MICs were interpreted following EUCAST guidelines (Supplementary Table 2). After completing these diagnostic steps, five to ten CFUs of each isolate were stored at −80 °C for future use.

### Real-time genomic data generation

The stored isolates were thawed and grown overnight at 37 °C on BD® Columbia blood agar plates. In the first round of sequencing, we isolated DNA from ten CFUs of the pre-and post-treatment isolates using an automated magnetic-bead-based DNA purification approach through the Maxwell® RSC Blood DNA extraction protocol for Gram-negative bacteria with the Promega Maxwell®RSC 48 Instrument (Promega Corporation, Madison, USA). DNA concentrations were measured using the Qubit™ (Thermo Fischer Scientific, Waltham, USA) dsDNA HS kit according to the manufacturer's instructions. Nanopore sequencing libraries of both samples were generated using the SQK-RBK004 Rapid Barcoding Kit and sequenced on an Oxford Nanopore Technologies MinION MK1b device with R9.4.1 flow cells for 15 h[28]. For the second round of sequencing of the pre-treatment isolate, we subcultured the stored isolate again, generating a technical replicate of the pre-treatment isolate. We then extracted DNA from circa 50 CFUs and sequenced the extracted DNA for 8 h (Supplementary Fig. 1).

### Real-time genomic data analysis

An overview of the real-time genomic data analysis is presented in Supplementary Fig. 1. All computational analyses were conducted on a portable laptop with an 8 GB NVIDIA GeForce RTX 4070 GPU, 16 GB 5200 MHz RAM, and an Intel i7-13800H CPU with 14 cores and 20 threads.

The raw nanopore data was basecalled using Guppy v6.3.2, using the "High-accuracy" model (r9.4.1_450bps_hac). We used Porechop v0.2.3 (https://github.com/rrwick/Porechop) to trim the adapter sequences and filtered out low-quality reads (Q < 9) and short sequences (< 200 bp) using Nanofilt v2.8.0 (https://github.com/wdecoster/nanofilt). Sequencing summaries were generated using NanoStat v1.6.0 (https://github.com/wdecoster/nanostat)[30].

Subsequent analyses involved the EPI2ME Fastq Antimicrobial Resistance (v2023.04.26–1808834) workflow, which includes quality control of the filtered and trimmed reads, taxon identification via the WIMP (What's in My Pot; v2023.06.13-1865548) workflow, based on the NCBI RefSeq database and Centrifuge[31], and resistance gene identification using the Comprehensive Antibiotic Resistance Database (CARD)[16]. We retained only the resistance gene detections identified by the workflow's protein homolog model, which is the most conservative model of the Fastq Antimicrobial Resistance workflow[16], and excluded resistance genes with detection accuracy below 90% according to the protein homolog model to minimize the false positive rate.

Subsequently, we created de novo assemblies using Flye v2.9.1 (https://github.com/fenderglass/Flye)[10,28]. These were polished using Minimap2 v2.18[32] and Racon v1.5 (https://github.com/isovic/racon). We assessed assembly coverage using SAMtools depth v1.19.2 (https://github.com/samtools/samtools)[33]. We then analyzed our de novo assemblies using the Pathogenwatch v2.3.1[18] platform which integrates Kleborate[19] for *Klebsiella* species complex assignments and identification of acquired virulence factors and recognized resistance markers. We additionally used Pathogenwatch for core genome multilocus sequence typing (cgMLST)[18,19] of our assemblies based on the Life Identification Number (LIN) code scheme for the assignment of sublineages and clonal groups.

### Plasmid detection and annotation

We visualized our assembly graphs using Bandage v0.90[34] (Fig. 4), and identified the chromosomal and plasmid genomes. The plasmid functional annotation was done using MOB-suite v3.1.8 and visualized using the mobileOG-db[17] implemented in ProkSee[35] (Fig. 4). We used the MOB-typer modules from the Mob-suite program[22] to identify key mobilization genes (relaxase), origin of transfer (oriT), mate-pair formation (MPF).

To estimate plasmid copy-number, we calculated the ratio of plasmid replicon sequencing depth to the sequencing depth of the respective chromosomal contig[36]. To accurately estimate the normalized abundance of specific resistance genes per plasmid, we extracted contig-specific read IDs using SAMtools v1.19.2[33], retrieved the respective sequencing reads from the processed fastq files using SeqKit v2.8.0[37], and calculated the copy-number ratio of the resistance gene of interest in comparison with the most abundant resistance gene identified on the same plasmid.

### Reference center annotation

The post-treatment bacterial isolate was submitted to the German National Reference Center for Gram-negative bacteria (https://memiserf.medmikro.ruhr-uni-bochum.de/nrz), where our $bla_{KPC-14}$ gene variant was identified as a previously undetected CAZ-AVI-resistant KPC subtype with reduced carbapenem-hydrolysing activity using short-read whole-genome sequencing (Illumina MiSeq). This KPC subtype is now registered as $bla_{KPC-159}$ (NCBI sequence ID: OQ450354.1).

### Reporting summary

Further information on research design is available in the Nature Portfolio Reporting Summary linked to this article.

## Data availability

The nanopore sequencing data generated in this study have been deposited in the NCBI database (https://www.ncbi.nlm.nih.gov, SRA submission: PRJNA1041345).

## Code availability

All computational scripts are available https://github.com/Genomics4OneHealth/AMR_nanopore/releases/tag/v0.

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

## Acknowledgements

We thank the laboratory technicians and physicians at the Institute of Medical Microbiology at the Technical University Munich for their

support with phenotypic species identification, susceptibility testing, and sample storage and retrieval. We thank Sam Chorlton for helpful discussions on the BugSeq platform. We thank Niels Pfennigwerth from the German National Reference Center for Gram-Negative Bacteria for his work on identifying the novel KPC-159 subtype. We thank Dr. Youssef Hamway for his help with laboratory work and conceptualization of the study. This project was funded by a Helmholtz Principal Investigator Grant awarded to L.U. DNA extraction was funded by C.P.dC.'s DZIF grant TTU 03.818_01, and initial clinical work by C.P.dC.'s Bavarian Ministry of Commerce grant 47-6665 g/1311/2-MW-2012-0008. Figures 1, 2 and Supplementary Fig. 1 were created with Biorender.com, Fig. 4 was created with ProkSee.com and Bandage.

## Author contributions

E.S. and L.U. conceptualised the project. E.S., L.U. and E.F.N. designed the experiments. E.S., T.R., and A.P. conducted the real-time sequencing experiments under L.U.'s supervision. E.S., T.R., and S.V. conducted the bioinformatic analysis including data curation, formal analysis, and visualisation, under L.U.'s and E.F.N.'s supervision. N.G. consulted on genomic data analysis and interpretation and edited manuscript drafts. N.W., and C.P.dC. provided samples and laboratory equipment for phenotypic testing. C.C. supervised and synthesized the clinical case, as well as the routine and reference laboratory diagnostics. E.S. and L.U. wrote the manuscript with input from all co-authors.

## Funding

## Competing interests

The authors declare no competing interests.
