## [Peer review file · Nature Communications]

Detection of hidden antibiotic resistance through real-time genomicsREVIEWER COMMENTS

Reviewer #1 (Remarks to the Author):

The authors have reported the utility of Nanopore technology in investigating the unexpected deterioration of a patient and proposed a how using this technology in real-time could have helped to manage the case. The report is well written, and figures are informative. The authors report the proposed times that it would take to generate the results and a new version of the KPC allele. This work is contributing to the overall improvement of the field and contributes as a well-needed case-study for genomic sequencing implementation into the routine clinical care, however, taking into account the increasing number of studies using Nanopore for AMR detection in real time, the current study lacks the novelty and direct impact, for which the expectation was set out by a well-written and catchy title (I was expecting to see something like metagenomic samples sequenced and resistance then detected in follow on samples in real-time, or multiple bacterial samples sequenced and overall impact has been shown).

There are couple suggestions and questions that would help to improve the report.

1. In the main text it is not straight away clear that you have been working with single colony purified samples after the samples have undergone routine microbiology procedures. And it is not clear that the authors have sequenced single or multi-colony samples?
2. There are no mention of limitations of not doing a metagenomic sequencing on the patients samples to see if multiple types of *Klebsiella pneumoniae* or even other bacterial pathogens could have contributed to the change in AST results. Would such isolates be available?
3. Was a blood sample available for the patient in the first days or only the Endo aspirate was taken?

4. “Using this real-time genomic data, we correctly identified *K. pneumoniae* as the dominant pathogen in the pre- and post-treatment isolates.” This sentence makes it seem that you have not expected to see *K. pneumoniae* or were expecting multiple pathogens? This unfortunately adds to the confusion if only pure known-pathogen colonies were sequenced, or you have sequenced a streak of bacterial growth.

5. Would authors please clarify for having different 15hr and 20hrs cut-offs for pre- and post-treatment isolates?

6. Table 1. I have failed to find the explanation in the methods for how the accuracy was estimated in these cases? What did you use as your standards for comparison? It did not seem that the authors run biological and technical replicates of the same samples to confirm that the levels of detection were the same, especially in the situation where they were not necessary following a well-established and tested methods for such scenarios (and its expected given that the field is rapidly evolving!). but having genomic data from biological and technical replicates – would have provided stronger evidence for if the KPC-2 and KPC-159/KPC-14 strains were co-existing already but were failed to be detected in time due to ratios of these mini-sub-populations present.

7. More information and illustration of genes identified on the detected IncN plasmid would have been very informative – was this predicted to be mobilizable plasmid? Did it have all conjugation mechanism in place in both pre- and post- treatment samples? How did genetic contents of the plasmid carrying KPC-2 or KPC-14/KPC-159 compared? This would provide additional information that could explain the potential advantage of KPC-14/KPC-159 emergence in such short time or at least contribute to the better understanding on IncN plasmids.

8. How many copies of plasmid replicons were found in comparison to copy numbers of the targeted genes? were there also any other resistance genes detected?

9. “a new KPC resistance gene, KPC-159, that is very similar to the KPC-14 gene detected by our analysis (amino acid sequence similarity of 99.9%) and therefore conferred the same

phenotypic antibiotic resistance” “- this sentence from the authors is very confusing, as now it raises questions was is KPC-14 reported here or the KPC-159? or all 3 were identified? if the latter, then what was the reasoning to report only KPC-14 in Table 1?

10. A more detailed genetic description of the strains and the plasmids identified (and maybe even potential comparison to other Kp strains reported globally to carry the reported KPC-2, KPC-14) would contribute to the understanding how spread this strain could be?

Reviewer #2 (Remarks to the Author):

The use of Nanopore for rapid characterisation of bacteria and AMR predictions from blood stream infections (BSI) is not groundbreaking. Harris et al recently published work evaluating the performance of Nanopore for this purpose (Harris et al. *Microbiology spectrum* (2024). DOI: <https://doi.org/10.1128/spectrum.03065-23>). However, the novelty here lies in the real-time application of Nanopore. Yet, it remains uncertain if the authors achieved their intended objectives.

The authors highlight lingering concerns about the precision of real-time genomics in predicting phenotypic antibiotic resistance, especially when juxtaposed with established diagnostic approaches" - it would greatly benefit readers if these concerns were described alongside how this study addresses them. My concern is that the authors failed to identify the heterogeneity with regards to meropenem and CAZ-AVI resistance observed in this patient. This might be attributed to how isolates were chosen for sequencing (colony pick versus plate sweep), underscoring a limitation of isolate sequencing i.e. its inability to efficiently capture population heterogeneity among closely related bacteria within individual patients. Unfortunately, the authors haven't sufficiently addressed this limitation or proposed strategies for how they may be overcome.

The authors describe the discovery of a "new" KPC resistance gene, which would be better described as a new allele. The novelty of this discovery is debatable, given its 99.9% amino acid sequence similarity to KPC-14 and its similar resistance phenotype.

Initial sequencing took ~5 days, which raises questions about its comparative efficiency and cost-effectiveness against standard laboratory diagnostic procedures. Did the subsequent sequencing phase also take around five days? If so, did the cumulative time for characterising the infection's resistance profile approach ten days?

While the entry cost for Nanopore sequencing is markedly lower than that of other sequencing platforms, the computational infrastructure required for high-quality base calling in a timely manner can sometimes surpass the sequencing platform's cost. It would be advantageous for the authors to delineate the computational infrastructure employed for bioinformatics analysis and whether it might pose a barrier to adopting and integrating whole-genome sequencing (WGS) into routine clinical practice.

Comments from Reviewer 1:

“1. In the main text it is not straight away clear that you have been working with single colony purified samples after the samples have undergone routine microbiology procedures. And it is not clear that the authors have sequenced single or multi-colony samples?”

Response:

Thank you very much for pointing out that the colony characteristics are currently not clear to the reader. We took multi-colony samples of up to 50 colony-forming units (CFUs) for the real-time genomic analysis while the routine microbiology procedures were done on up to three colonies from multi-colony samples. First, please find below an excerpt from the Methods section on page 9 (ll. 245-256) which describes the routine microbiology diagnostics:

“Following subculture, species identification was done from a single colony forming unit (CFU) of pure bacterial isolates using Matrix-assisted laser desorption ionization time-of-flight mass spectrometry instructions (MALDI-TOF MS, Bruker Daltronics GmbH, Leipzig Germany), as per the manufacturer’s instructions.

Antibiotic susceptibility testing was performed using VITEK2 (BioMérieux, Marcy l’Etoile, France). For this, up to three bacterial CFUs were transferred to a saline tube to generate a homogenous suspension with a density equivalent to 0.5 McFarland. Subsequently, Minimum Inhibitory Concentrations (MICs in mg/L) of the two isolates were determined using the VITEK2 gram-negative (AST-GN69) card, and the results interpreted according to the European Committee on Antimicrobial Susceptibility Testing (EUCAST) guidelines²⁶ (**Supplementary Table 2**).”

After routine microbiology diagnostics, we stored multiple colonies from subcultures generated during routine diagnostics. These stored isolates were then subcultured for our genomic analyses, and up to 50 CFUs from these subcultures were used for DNA extractions.

This is now emphasized in the Methods section on page 10 (ll. 274-276):

“The stored isolates were thawed and grown overnight at 37°C on BD® Columbia blood agar plates. In the first round of sequencing, we isolated DNA from ten CFUs of the pre- and post-treatment isolates using an automated magnetic-bead-based DNA purification approach (...).”

Importantly, in order to clarify that we processed the isolates after routine microbiology procedures in the manuscript, we have additionally changed the following sentence on page 4 (ll. 117-121):

- from “We applied nanopore shotgun sequencing (Oxford Nanopore Technologies) to both *K. pneumoniae* isolates (pre- and post-CAZ-AVI treatment) by using the portable Mk1b sequencing device, rapid barcoding library preparation, and R9 nanopore chemistry (**Figure 2; Methods**).”
- to: “To explore the potential of real-time genomics in antibiotic resistance prediction, we simulated its application after completion of routine diagnostics for this clinical case as follows. We applied rapid nanopore shotgun sequencing (Oxford Nanopore Technologies) on both the pre- and post-treatment *K. pneumoniae* bacterial isolates (**Methods**) using the portable Mk1b sequencing device and rapid barcoding library preparation (**Figure 2; Supplementary Table 1; Methods**).”

In addition, we now clarify this in the figure caption of Figure 2 on page 4 (ll. 111 to 113):

“After completion of the routine diagnostics, we used real-time genomics to sequence DNA from the pre- and post-CAZ-AVI treatment bacterial isolates using the portable nanopore sequencing device Mk1b (Methods).”

“2. In addition, there are no mention of limitations of not doing a metagenomic sequencing on the patients samples to see if multiple types of *Klebsiella pneumoniae* or even other bacterial pathogens could have contributed to the change in AST results. Would such isolates be available?”

Response:

We fully agree with the reviewer that metagenomic sequencing of the original patient sample without bacterial culturing would provide the ideal data for future genomic assessments of infection-causing pathogens in the clinic – to increase the speed of diagnosis and to assess all potential pathogens independently of any culture bias.

First, our study has a different focus, which we have now made clearer throughout our manuscript. We here tried to elucidate the mechanisms of a “switch” of antibiotic resistance in a patient that could not be understood by established clinical diagnostics: As standard antibiotic resistance tests were not able to explain the antibiotic resistance profile of the patient’s infection, we used this example to show that long-read genomics – and specifically nanopore sequencing that can be employed *in situ* in the clinic and in real-time – would have been able to predict the “correct” antibiotic resistance patterns from rare plasmids that could have been detected at the time that the patient presented at the clinic. This application of real-time genomics therefore shows that phenotypically hidden resistance can be revealed by genomic analyses, with important consequences for the clinical management of this case (*i.e.*, the last-resort antibiotic CAZ-AVI would have not been administered). To the best of our knowledge, this is the first time that a discrepancy between currently established clinical approaches and treatment outcome could be fully elucidated through genomic data. **In order to directly compare those established diagnostics with real-time genomic predictions in this specific case, we believe that it has been important to apply both approaches to the same sample type, *i.e.* the bacterial culture.**

Second, – on a more practical note – we used an existing stock of isolated bacterial cultures stored at -80°C from this past infection case, so that at the time that we retrospectively conducted our genomic

study the original patient samples had already been discarded according to the routine diagnostic protocol in our microbiology laboratory. We agree that a possibility exists that another pathogen could have harbored the resistance genes responsible for the secondary antibiotic resistance in the patient, which neither established clinical nor genomic approaches could have detected from the *Klebsiella* culture. We, however, emphasize that this only applies to non-culturable bacteria or bacteria that were present at too low abundances to be cultured: Many other known pathogens would have also been isolated as bacterial cultures by routine diagnostics, and even if the bacterial culprit had been present in low concentrations in the first infection it would have been dominant enough in the second infection to be culturable. We therefore believe that we here present the most parsimonious explanation of all evidence in this case, especially given that only *Klebsiella* could be isolated and that both *Klebsiella* isolates contained the same resistance gene. We, however, fully agree with the reviewer that the **limitations of bacterial isolate sequencing in comparison to metagenomics should be highlighted in our manuscript**. We therefore now emphasize this in our Discussion, and mention the potential possibility of another non-culturable bacteria causing the secondary resistance (page 8, ll. 217-226):

“While our analysis provides evidence for the added value of real-time genomics for complex bacterial infections, two significant limitations to our work remain. Firstly, focusing on bacterial isolates may have limited our ability to fully understand the microbial diversity of the patient’s infection. A metagenomic sequencing approach using direct patient samples, rather than cultured bacteria, could have identified non-culturable organisms that might have influenced the antibiotic resistance profile⁷. Secondly, our analysis was confined to just two bacterial isolates from the same patient over an extended period. A larger sample size and more frequent sampling might have provided more detailed insights into how plasmid selection evolved in response to the antibiotic selection pressure. Unfortunately, due to the retrospective nature of this study, we were unable to adjust the sample size or obtain direct patient samples.”

Was a blood sample available for the patient in the first days or only the Endo aspirate was taken?” Response:

The primary endotracheal aspirate and the secondary blood culture were the only samples from which multi-drug resistant *Klebsiella pneumoniae* could be isolated. Any prior blood cultures showed no bacterial growth, and all original patient samples had been discarded at the time of this retrospective study.

“Using this real-time genomic data, we correctly identified *K. pneumoniae* as the dominant pathogen in the pre- and post-treatment isolates.” This sentence makes it seem that you have not expected to see *K. pneumoniae* or were expecting multiple pathogens? This unfortunately adds to the confusion if only pure known-pathogen colonies were sequenced, or you have sequenced a streak of bacterial growth.”

Response:

We changed this sentence to the following to clarify that what we meant was that the real-time genomics approach predicted the same putative pathogen as established diagnostics (page 4, ll. 115 to 116): “We correctly identified *K. pneumoniae* in the pre- and post-treatment isolates as the putative pathogen.”.

Would authors please clarify for having different 15hr and 20hrs cut-offs for pre- and post-treatment isolates?”

Response:

The difference in sequencing run time was purely due to logistical reasons, but we agree that the different sequencing run times might be confusing in the current version of the manuscript. The first and second isolate were sequenced for 15h to ensure that sufficient genomic data was produced (>100 Mb). As the second sequencing run of the first sample (the “+20h” run) was only done to simulate the potentially adaptive nature of nanopore sequencing in the clinic (i.e. the ability to sequence as long as necessary until conclusive results are obtained through real-time analysis), we can restrict the data analysis to the first 8h of sequencing when five highly accurate KPC-14 copies had been obtained. We now added Figure 3 on page 6 (ll. 240-246) to the manuscript to illustrate the cumulative read output and KPC-14 copy-number detection over sequencing time.

Table 1. I have failed to find the explanation in the methods for how the accuracy was estimated in these cases? What did you use as your standards for comparison? It did not seem that the authors run biological and technical replicates of the same samples to confirm that the levels of detection were the same, especially in the situation where they were not necessary following a well-established and tested methods for such scenarios (and its expected given that the field is rapidly evolving!). but having genomic data from biological and technical replicates – would have provided stronger evidence for if the KPC-2 and KPC-159/KPC-14 strains were co-existing already but were failed to be detected in time due to ratios of these mini-sub-populations present.”

Response:

The accuracy in Table 1 refers to the accuracy as measured by the CARD protein homolog resistance detection model, *i.e.* it represents the average alignment accuracy across all sequencing reads to the respective CARD AMR reference gene. We now clarify this in the table caption on page 5, l. 135: “Accuracy refers to gene detection accuracy according to the protein homolog model.”

We further fully agree with the reviewer that replicates could have strengthened our study. We were not able to obtain biological replicates since the frozen bacterial isolates we used were the only remaining samples from this clinical diagnostics case. In order to, however, show that we did not only sequence a snapshot of the original bacterial culture, we re-cultured the stored sample from the same stock for the second sequencing round three times and extracted DNA from the pooled culture of at least 50 CFUs, thus generating technical replicates of the first sequencing results. We now explain this in more detail in the Methods section (page 10, ll. 282- 285):

“For the second round of sequencing of the pre-treatment isolate, we subcultured the stored isolate again, generating a technical replicate of the pre-treatment isolate. We now extracted DNA from circa 50 CFUs and sequenced the extracted DNA for 8 hours (**Supplementary Figure 1**).”

Following the reviewer’s comment, we further decided to create an additional sequencing replicate to test the robustness of our results: We therefore started another nanopore sequencing run of the same DNA extract of the second sequencing round of the pre-treatment isolate, following the same sequencing protocol and data analysis approach. Within 22h of sequencing, we obtained 1.6 Gb of data with an average Phred quality score of 11.6 and an average read length of 5,816 bases. We

detected six copies of KPC-14 with an average alignment accuracy of 90.2% and 152 copies of KPC-2 with an average alignment accuracy of 92.8%.

We would be very happy to provide the respective fastq file of this technical replicate to the reviewers. Please let us know if and how you would like us to transfer this additional data.

More information and illustration of genes identified on the detected IncN plasmid would have been very informative – was this predicted to be mobilizable plasmid? Did it have all conjugation mechanism in place in both pre- and post- treatment samples? How did genetic contents of the plasmid carrying KPC-2 or KPC-14/KPC-159 compared? This would provide additional information that could explain the potential advantage of KPC-14/KPC-159 emergence in such short time or at least contribute to the better understanding on IncN plasmids.”

Response:

Thank you for this very valuable comment. We agree that more functional analyses of the IncN plasmids in the pre- and post-treatment samples can be valuable and we now therefore provide additional information on functional characteristics of the plasmids in our new version of the manuscript. In general, we argue that the main reason for the emergence of the KPC-14/KPC-159 gene carrying plasmid was due to the selection pressure under CAZ-AVI exposure. We now analyzed the plasmid functionality with Mob-suite¹ and additionally visualized the assemblies and the functional annotation using the online tool ProkSee (<https://proksee.ca/>) (see Figure 4, page 7). In summary, the plasmids harbored the same relaxase type (MOBF), mpf type (MPF_T), andorit type (MOBF), and were both predicted to be conjugative. We summarized these results in the revised manuscript as follows, and visualized the results in an extended Figure 4 (page 7):

Results page 6 (ll. 167-171):

“Functional annotation of the assembled plasmid genomes revealed that the KPC-2 (pre-treatment isolate) and KPC-14 (post-treatment isolate) were both located on IncN plasmids, which were 99.7%-identical according to sequence alignments (E-score 0, Bit-Score>1.461e+5; **Methods; Figure 4**). Additionally, both IncN plasmids shared key plasmid features (relaxase type: MOBF, mpf type: MPF_T, orit type: MOBF), and were predicted to be conjugative^{21,22}.”

Methods page 11 (ll. 318-321):

“The plasmid functional annotation was done using MOB-suite¹⁸ and visualized using the mobileOG-db¹⁷ implemented in ProkSee³⁷. We used the MOB-typer modules from the Mob-suite program³⁸ to identify key mobilization genes (relaxase), origin of transfer (oriT), mate-pair formation (MPF).”

How many copies of plasmid replicons were found in comparison to copy numbers of the targeted genes? Were there also any other resistance genes detected?”

Response:

In the revised manuscript, we now compare the sequencing depth of plasmid replicons to the sequencing depth of chromosomal genes. This comparison helps us estimate how many copies of

plasmids are present in relation to the chromosomal DNA (similar to previous work by Wick et al.²). To provide a more in-depth analysis of plasmid abundance beyond what differential contig coverage can suggest, we additionally assessed the resistance gene copy-number per contig for all contigs that were identified as IncN plasmids. We then estimated the normalized abundance of KPC-14 carrying plasmids based on the copy-number of the most prevalent resistance gene (Tem-4) identified in all isolates. We added this analysis to the following sections of the revised manuscript:

Results page 6 (ll. 173-178):

“We further inferred a copy-number of three and four for the IncN plasmids relative to the bacterial chromosome in the pre- and post-treatment isolates, respectively (**Methods**). To assess the changes in KPC-14 abundance between pre- and post-treatment isolates, we further normalized the KPC-14 copy-numbers by the copy-number of the most abundant resistance gene (TEM-4) detected on the IncN plasmid (**Methods**); we found that the normalized abundance of KPC-14 increased from 0.56% to 26.6% after CAZ-AVI exposure.”

Methods page 11 (ll. 323-328):

“To estimate plasmid copy-number, we calculated the ratio of plasmid replicon sequencing depth to the sequencing depth of the respective chromosomal contig³⁹. To accurately estimate the normalized abundance of specific resistance genes per plasmid, we extracted contig-specific read IDs using SAMtools v1.19.2³⁴, retrieved the respective sequencing reads from the processed fastq files using SeqKit v2.8.0⁴⁰, and calculated the copy-number ratio of the resistance gene of interest in comparison with the most abundant resistance gene identified on the same plasmid.”

In reference to the reviewer’s second question, we were indeed able to detect additional resistance genes which all corroborated the phenotypic resistance results from routine diagnostics. As our study focuses on the clinically relevant and phenotypically hidden CAZ-AVI resistance, we limited our results in the manuscript to resistance genes conferring resistance to this last-resort antibiotics. We, however, appreciate that the reader might be interested in all detected resistance genes in all samples, which we have now attached as Supplementary Table 3 to the manuscript. We have further made sure that all data and computational scripts are available via open-access to fully replicate the analyses.

“9. “a new KPC resistance gene, KPC-159, that is very similar to the KPC-14 gene detected by our analysis (amino acid sequence similarity of 99.9%) and therefore conferred the same phenotypic antibiotic resistance” “- this sentence from the authors is very confusing, as now it raises questions was is KPC-14 reported here or the KPC-159? or all 3 were identified? if the latter, then what was the reasoning to report only KPC-14 in Table 1?”

Response:

We thank the reviewer for highlighting this potential source of confusion. Our study is the first report of this new KPC-159 subtype, whose genomic reference sequence we subsequently added to the NCBI database. However, since our resistance gene detection relied on the existing CARD database which has not yet been extended by this newly discovered KPC subtype, our analyses only identified and reported the closest genetic match of KPC-159, namely KPC-14, with a sequence similarity of 99.9%.

We now further clarify this in our Results section (page 7, ll. 188-193):

“Upon submission of the post-treatment bacterial isolate to the German National Reference Center for Gram-negative bacteria, the KPC resistance gene that we initially defined as KPC-14 was identified as a previously undocumented KPC subtype and subsequently named *KPC-159* (NCBI sequence ID: OQ450354.1). To confirm this, we utilised the BLASTn tool and established that KPC-159 shows 99.9% similarity in protein sequence (875/876 amino acids) with KPC-14 (**Methods**), thus, leading to the classification of the KPC-149 as KPC-14 by the protein homolog model.”

as well as in our Methods section (page 21, ll. 331- 335):

“The post-treatment bacterial isolate was submitted to the German National Reference Center for Gram-negative bacteria (<https://memiserf.medmikro.ruhr-uni-bochum.de/nrz>), where the KPC-14 gene variant was identified as a previously undetected CAZ-AVI-resistant KPC subtype with reduced carbapenemase activity using short-read whole-genome sequencing (Illumina MiSeq). This KPC subtype is now registered as *KPC-159* (NCBI sequence ID: OQ450354.1).”

“10. A more detailed genetic description of the strains and the plasmids identified (and maybe even potential comparison to other Kp strains reported globally to carry the reported KPC-2, KPC-14) would contribute to the understanding how spread this strain could be?”

Response:

We followed up on the reviewer’s suggestion and utilized the global collection of *K. pneumoniae* strains in Pathogenwatch, which includes 48,005 genomes as of March 15th 2024, along with its single-linkage clustering tool based on the LIN code scheme. Our findings indicate that our strains are genetically distinct when compared to other publicly available genomes. We included these results in the Results section (page 6, ll. 161-165):

“Core-genome multilocus sequence typing (cgMLST; **Methods**)^{18,19} revealed that both isolates were of the emerging high-risk sequence type ST147²⁰. Single-linkage clustering analysis identified no close relatives within the 50-allele threshold typically used for cgMLST clustering (**Methods**), indicating that our bacterial isolates are genetically distinct from globally known ST147 genomes.”

and in the Methods section (page 11, ll. 310- 314):

“We then analyzed our *de novo* assemblies using the Pathogenwatch v2.3.1¹⁸ platform which integrates Kleborate¹⁹ for *Klebsiella* species complex assignments and identification of acquired virulence factors and recognized resistance markers. We additionally used Pathogenwatch for core genome multi-locus sequence typing (cgMLST)^{18,35} of our assemblies based on the Life Identification Number (LIN) code scheme for the assignment of sublineages and clonal groups.”

Our review of the existing literature on IncN plasmids revealed no evidence of a correlation between IncN plasmids, KPC-14, and ST147. In Germany, the predominant resistance genes found on IncN plasmids were linked to KPC-2³. Similarly, most studies on ST147 highlighted associations with KPC-2, NDM, and VIM plasmids⁴, suggesting a different resistance profile than in our post-treatment isolate. Literature specifically addressing KPC-14 carrying plasmids is scarce, with only two studies reporting its presence on a specific plasmid: One study identified KPC-14 on an IncN plasmid within ST16⁵, and another found it on an IncFII/IncR plasmid in ST11⁶. Unfortunately, the lack of sequencing data for KPC-14 plasmids made further analysis of plasmid clustering impossible.

Main reviewer 1 assessment:

“This work is contributing to the overall improvement of the field and contributes as a well-needed case-study for genomic sequencing implementation into the routine clinical care, however, taking into account the increasing number of studies using Nanopore for AMR detection in real time, the current study lacks the novelty and direct impact, for which the expectation was set out by a well-written and catchy title (I was expecting to see something like metagenomic samples sequenced and resistance then detected in follow on samples in real-time, or multiple bacterial samples sequenced and overall impact has been shown).”

Response:

We hope to have addressed the limitation pointed out by reviewer 1 in our revised manuscript, and to have highlighted the significance of our findings more clearly. Briefly, to the best of our knowledge, this is the first time that a discrepancy between currently established clinical approaches and treatment outcome could be fully elucidated through genomic data: We showed that real-time genomics could elucidate the mechanisms of a “switch” of antibiotic resistance in a patient that could not be understood by established clinical diagnostics; the *in situ* and real-time application of nanopore sequencing in the clinic would have therefore been able to predict the “correct” antibiotic resistance patterns from rare plasmids that could have been detected at the time that the patient presented at the clinic. This application of real-time genomics shows that phenotypically hidden resistance can be revealed by genomic analyses, with potentially important consequences for the clinical management. We additionally showed that we could correctly predict the resistance profile from genomic data while it was conferred by a previously unseen resistance gene subtype (KPC159) with a rarely observed increased avibactam-hydrolyzing activity and decreased carbapenem-hydrolyzing activity; this illustrates that nanopore sequencing data can uncover novel resistance mechanisms, enhancing our understanding of bacterial resistance and guiding treatment and infection prevention protocols in respective clinical cases.

Comments from Reviewer 2

“1. The use of Nanopore for rapid characterisation of bacteria and AMR predictions from blood stream infections (BSI) is not groundbreaking. Harris et al recently published work evaluating the performance of Nanopore for this purpose (Harris et al. Microbiology spectrum (2024). DOI: <https://doi.org/10.1128/spectrum.03065-23>). However, the novelty here lies in the real-time application of Nanopore. Yet, it remains uncertain if the authors achieved their intended objectives.”

Response:

We fully acknowledge the breadth of research in antimicrobial resistance predictions using nanopore sequencing, but we argue that our work introduces a completely novel aspect: To the best of our knowledge, this is the **first time that a discrepancy between currently established clinical diagnostics and treatment outcome could be fully elucidated through genomic data**: We showed that real-time genomics could elucidate the mechanisms of a “switch” of antibiotic resistance in a patient that could not be understood by established clinical diagnostics; the *in situ* and real-time application of nanopore sequencing in the clinic would have therefore been able to predict the “correct” antibiotic resistance patterns from rare plasmids that could have been detected at the time that the patient presented at the clinic. This application of real-time genomics shows that phenotypically hidden resistance can be revealed by genomic analyses, with potentially important consequences for the clinical management. **We have now made this unique aspect of our study clearer throughout the manuscript.**

We further added Harris *et al.* (2024), which was published after our initial submission, as a reference and highlighted the relevance of already existing nanopore sequencing-based clinical studies on page 2 (ll. 51-54):

“While several proof-of-concept studies have showcased the feasibility of using nanopore sequencing for rapid infectious disease diagnostics in clinical settings⁵⁻⁹, it remains to be proven that real-time genomics can outperform established diagnostics in detecting clinically relevant resistance.”

The authors highlight lingering concerns about the precision of real-time genomics in predicting phenotypic antibiotic resistance, especially when juxtaposed with established diagnostic approaches" - it would greatly benefit readers if these concerns were described alongside how this study addresses them. My concern is that the authors failed to identify the heterogeneity with regards to meropenem and CAZ-AVI resistance observed in this patient. This might be attributed to how isolates were chosen for sequencing (colony pick versus plate sweep), underscoring a limitation of isolate sequencing i.e. its inability to efficiently capture population heterogeneity among closely related bacteria within individual patients. Unfortunately, the authors haven't sufficiently addressed this limitation or proposed strategies for how they may be overcome.”

Response:

Thank you for highlighting this issue. We agree with the concern raised about our study's limitation to sequencing bacterial isolates rather than actual patient samples. We have answered this concern in our response to the first reviewer's question number (2.). Please let us know if there is anything else to clarify.

The authors describe the discovery of a "new" KPC resistance gene, which would be better described as a new allele. The novelty of this discovery is debatable, given its 99.9% amino acid sequence similarity to KPC-14 and its similar resistance phenotype.”

Response:

As requested, we have updated our terminology, refraining from calling the KPC-159 subtype a novel “gene”, given the high sequence similarity between KPC-14 and KPC-159 (NCBI sequence ID: OQ450354.1). Additionally, we would like to clarify that the discovery of this novel KPC allele is not the main finding of this study; the novelty instead lies in demonstrating that nanopore sequencing can reliably predict phenotypically hidden resistance patterns, including those of previously unidentified resistance alleles which are only present on low-abundance plasmids. We now emphasize this focus more throughout our revised manuscript.

Initial sequencing took ~5 days, which raises questions about its comparative efficiency and cost-effectiveness against standard laboratory diagnostic procedures. Did the subsequent sequencing phase also take around five days? If so, did the cumulative time for characterising the infection's resistance profile approach ten days?”

Response:

We only applied nanopore sequencing for 15h, followed by up to 8h of sequencing to simulate the adaptive application of nanopore sequencing in the clinic. While we are not certain where this

misunderstanding came from, we have double-checked all methodological descriptions in the revised version of our manuscript and believe that all details of our protocol are clear now.

“5. While the entry cost for Nanopore sequencing is markedly lower than that of other sequencing platforms, the computational infrastructure required for high-quality base calling in a timely manner can sometimes surpass the sequencing platform's cost. It would be advantageous for the authors to delineate the computational infrastructure employed for bioinformatics analysis and whether it might pose a barrier to adopting and integrating whole-genome sequencing (WGS) into routine clinical practice.”

Response:

Thank you for pointing out that we have not detailed the computational requirements. High-accuracy basecalling can easily be implemented on portable and cost-efficient computational devices such as laptops with powerful GPUs (<USD 2k) or embedded AI machines from NVIDIA (<USD 1k). We now added all the computational specifics of the portable laptop we used for basecalling and *de novo* genome assembly to the manuscript (page 10, ll. 288-290, and figure caption of Supplementary Figure 1):

“All computational analyses were conducted on a portable laptop with an 8 GB NVIDIA GeForce RTX 4070 GPU, 16 GB 5200 MHz RAM, and an Intel i7-13800H CPU with 14 cores and 20 threads.”

We would again like to thank the editor and reviewers for the very helpful comments, and look forward to receiving further feedback.

With best regards,
Ela Sauerborn & Lara Urban, on behalf of all authors

References

1. Robertson, J. & Nash, J. H. E. MOB-suite: software tools for clustering, reconstruction and typing of plasmids from draft assemblies. *Microb Genom* **4**, (2018).
2. Wick, R. R., Judd, L. M., Wyres, K. L. & Holt, K. E. Recovery of small plasmid sequences via Oxford Nanopore sequencing. *Microb Genom* **7**, (2021).
3. Yao, Y. *et al.* Predominant transmission of KPC-2 carbapenemase in Germany by a unique IncN plasmid variant harboring a novel non-transposable element (NTE_{KPC-Y}). *Microbiol Spectr* **12**, (2024).
4. Peirano, G., Chen, L., Kreiswirth, B. N. & Pitout, J. D. D. Emerging Antimicrobial-Resistant High-Risk Klebsiella pneumoniae Clones ST307 and ST147. *Antimicrob Agents Chemother* **64**, (2020).
5. Niu, S. *et al.* A Ceftazidime-Avibactam-Resistant and Carbapenem-Susceptible Klebsiella pneumoniae Strain Harboring *bla*_{KPC-14} Isolated in New York City. *mSphere* **5**, (2020).
6. Wang, L., Shen, W. & Cai, J. Mobilization of the blaKPC-14 gene among heterogenous plasmids in extensively drug-resistant hypervirulent Klebsiella pneumoniae. *Front Microbiol* **14**, (2023).

REVIEWERS' COMMENTS

Reviewer #1 (Remarks to the Author):

I am happy with the edits and additional experimental work carried out by the authors, the manuscript has greatly improved. One very minor note would be to check if it is correct to use "putative pathogen" in sentence "Our analysis correctly identified *K. pneumoniae* in the pre- and post-treatment isolates as the putative pathogen" (page 5, 123-124) feels that it should be here "causative pathogen".

Comments from Reviewer 1:

I am happy with the edits and additional experimental work carried out by the authors, the manuscript has greatly improved. One very minor note would be to check if it is correct to use "putative pathogen" in sentence "Our analysis correctly identified *K. pneumoniae* in the pre- and post-treatment isolates as the putative pathogen" (page 5, 123-124) feels that it should be here "causative pathogen".

Response:

We have now changed the wording from "Our analysis correctly identified *K. pneumoniae* in the pre- and post-treatment isolates as the putative pathogen" (page 5, 123-124) to "Our analysis correctly identified *K. pneumoniae* in the pre- and post-treatment isolates as the **causative** pathogen".

Comments from Reviewer 2:

No comments given.